# A Comparison between Conventional and Extracorporeal Cardiopulmonary Resuscitation in Out-of-Hospital Cardiac Arrest: A Systematic Review and Meta-Analysis

**DOI:** 10.3390/healthcare10030591

**Published:** 2022-03-21

**Authors:** Reem Alfalasi, Jessica Downing, Stephanie Cardona, Bobbi-Jo Lowie, Matthew Fairchild, Caleb Chan, Elizabeth Powell, Ali Pourmand, Alison Grazioli, Quincy K. Tran

**Affiliations:** 1Department of Critical Care Medicine, New York Presbyterian Hospital/Columbia University Irving Medical Center, New York, NY 10032, USA; 2Program in Trauma, The R Adams Cowley Shock Trauma Center, University of Maryland School of Medicine, Baltimore, MD 21201, USA; jessica.downing@umm.edu (J.D.); elizabeth.powell@som.umaryland.edu (E.P.); agrazioli@som.umaryland.edu (A.G.); qtran@som.umaryland.edu (Q.K.T.); 3Department of Critical Care Medicine, Mount Sinai Hospital, New York, NY 10029, USA; stephanie.cardona@mountsinai.org; 4Department of Emergency Medicine, University of Maryland School of Medicine, Baltimore, MD 21201, USA; blowie@som.umaryland.edu; 5Research Associate Program in Emergency and Critical Care, Department of Emergency Medicine, University of Maryland School of Medicine, Baltimore, MD 21201, USA; matthew.fairchild@umm.edu; 6Division of Pulmonary and Critical Care Medicine, University of Maryland School of Medicine, Baltimore, MD 21201, USA; cchan@som.umaryland.edu; 7Department of Emergency Medicine, George Washington University School of Medicine and Health Sciences, Washington, DC 20052, USA; pourmand@gwu.edu

**Keywords:** ECMO, VA ECMO, extracorporeal membrane oxygenation, OHCA, out-of-hospital cardiac arrest, survival to discharge, neurologic outcome

## Abstract

There is limited evidence comparing the use of extracorporeal cardiopulmonary resuscitation (ECPR) to CPR in the management of refractory out-of-hospital cardiac arrest (OHCA). We conducted a systematic review and meta-analysis to compare survival and neurologic outcomes associated with ECPR versus CPR in the management of OHCA. We searched PubMed, EMBASE, and Scopus to identify observational studies and randomized controlled trials comparing ECPR and CPR. We used the Newcastle–Ottawa Scale and Cochrane’s risk-of-bias tool to assess studies’ quality. We used random-effects models to compare outcomes between the pooled populations and moderator analysis to identify sources of heterogeneity and perform subgroup analysis. We identified 2088 articles and included 13, with 18,620 patients with OHCA. A total of 16,701 received CPR and 1919 received ECPR. Compared with CPR, ECPR was associated with higher odds of achieving favorable neurologic outcomes at 3 (OR 5, 95% CI 1.90–13.1, *p* < 0.01) and 6 months (OR 4.44, 95% CI 2.3–8.5, *p* < 0.01). We did not find a significant survival benefit or impact on neurologic outcomes at hospital discharge or 1 month following arrest. ECPR is a promising but resource-intensive intervention with the potential to improve long-term outcomes among patients with OHCA.

## 1. Introduction

Out-of-hospital cardiac arrest (OHCA) is a leading cause of morbidity and mortality in the United States [1]. In 2018, over 74.3 per 100,000 individuals in the United States experienced OHCA and the incidence of OHCA treated by emergency medical services (EMS) personnel has been increasing over time [1]. Most episodes of sudden cardiac arrest occur in private residences and are unwitnessed; less than 40% receive bystander cardiopulmonary resuscitation (CPR) [1]. It has been well established that high-quality bystander CPR and early defibrillation for shockable rhythms lead to improved outcomes; despite this, almost 90% of individuals with sudden cardiac arrest will not survive [2,3,4]. Rates of survival to hospital discharge have previously been estimated at 9.9–10.4%, with only 8% of patients surviving with a favorable neurologic outcome [1,2,3,4]. Survival to hospital discharge is even lower for patients with refractory cardiac arrest, that is, patients requiring CPR for greater than 30 min: approximately 1%, 0.4% with a favorable neurologic outcome [5].

The advent of venoarterial extracorporeal membrane oxygenation (VA ECMO) has allowed for continued treatment and life-saving attempts in patients with refractory cardiac arrest [6]. Extracorporeal cardiopulmonary resuscitation (ECPR) is defined as the use of VA ECMO in the context of ongoing refractory cardiac arrest. A large-bore catheter is placed centrally or peripherally in the venous circulation to drain deoxygenated blood, which enters a peripherally placed device for oxygenation [6]. The oxygenated blood is then returned to the patient through a second peripheral or centrally placed catheter in the arterial circulation; a pump within the circuit supports circulation [6]. The ECMO circuit thus provides external cardiopulmonary support, allowing for perfusion of vital organs in the absence of a native heartbeat and serving as a bridge to additional therapies or recovery. ECMO is a highly resource-intensive and costly treatment strategy with limited availability in most hospital systems. Any widespread adoption of ECPR would require significant investment, as well as retraining and restructuring prehospital care systems to allow for transport to ECPR capable centers.

The recent ARREST trial—a small, phase 2, single-center, open-label, adaptive, safety and efficacy randomized controlled trial (RCT), and the only RCT to date directly comparing ECPR and CPR—found a higher survival rate in the ECPR group (43%) in comparison to the CPR group (7%) [7]. This study utilized rigorous inclusion and exclusion criteria; patients needed to have a shockable rhythm, no ROSC after three defibrillation shocks, and an estimated transfer time of less than 30 min. These criteria ensured that patients had minimal intervals of no-flow time. The strict exclusion criteria of trauma and presence of multiple comorbidities also aimed to exclude patients with low likelihood of survival. We observed that strict selection criteria for ECPR can be associated with improved patient outcomes compared to CPR. Despite this promising evidence, the ARREST trial consisted of a small sample size with highly trained personnel, and it is unclear if their result can be generalized outside this setting and population. ECPR should not replace conventional CPR until there are more clear and definitive criteria for ECPR; prompt and high-quality CPR remains the main effort to optimize patients’ outcomes.

We recently conducted a meta-analysis that identified a survival to hospital discharge rate of 24%, and 18% rate of survival with favorable neurologic outcomes, among patients with OHCA treated with ECPR [8]. Although these outcomes are improved when compared to those previously reported among patients with refractory OHCA treated with CPR, a direct comparison of the two therapies is still required to quantify the benefits [5,9]. As such, further research is required to determine benefits of ECPR in OHCA with respect to both survival and neurologic outcomes. This systematic review and meta-analysis assesses the current literature comparing survival and neurologic outcomes among patients treated with ECPR or CPR in OHCA.

## 2. Methods

### 2.1. Search Selection and Selection Criteria

We performed this study in accordance with the 2020 Preferred Reporting Items for Systematic-Review and Meta-Analysis (PRISMA) statement [10]. We conducted searches of EMBASE, PubMed, and SCOPUS on 6/10/2020 and 5/8/2021 to identify studies that compared the use of CPR and ECPR in OHCA. We used the search terms (pre-hospital) OR (out-of-hospital) AND (cardiopulmonary resuscitation) OR (cardiopulmonary arrest) OR (cardiac arrest) AND (ECPR) OR (extracorporeal cardiopulmonary resuscitation) OR (extracorporeal support) OR (extracorporeal membrane oxygenation). Studies were only included in our analysis if they reported survival or neurologic outcomes at or following hospital discharge or 30 days after arrest in adult patients (≥18 years) who experienced OHCA and included both CPR and ECPR treatment groups. We included retrospective and prospective observational studies with matched or unmatched cohorts, as well as randomized controlled trials. We excluded studies that: (1) did not report outcomes for OHCA patients separately (i.e., reported outcomes only for in-hospital cardiac arrest (IHCA) or did not distinguish outcomes for IHCA and OHCA), (2) included pediatric patients only, or did not differentiate outcomes between pediatric and adult patients, (3) focused on patients suffering specifically hypothermic cardiac arrests (prior studies have already demonstrated a benefit associated with ECMO in these cases, thought to be due to the stabilization of hemodynamics and allowance of time for rewarming) [11,12], (4) included non-human subjects, (5) were published in languages other than English, or (6) were case reports or case series, meeting abstracts, poster presentations, or other systematic reviews and meta-analyses.

We reviewed the enrollment years and locations (including registries and hospitals) of all eligible studies to identify potentially duplicated data. Studies potentially overlapping patients but that reported different outcomes of interest were not excluded. To manage the studies included in our meta-analysis, we used Covidence (www.covidence.org, accessed on 25 January 2022). Each title and abstract was independently reviewed by two investigators, and any disagreements were resolved by a third senior investigator. The same process was subsequently utilized for full text screening.

### 2.2. Outcomes

The primary outcome of interest was any favorable outcome, a composite outcome of: (1) survival to hospital discharge, (2) favorable neurologic outcome at hospital discharge or (3) favorable neurologic outcome at 1, 3, or 6 months after arrest. This composite outcome allowed us to utilize the primary outcome reported by the authors in each included study. Our secondary outcomes were survival to hospital discharge and neurologic outcome at hospital discharge and at 1, 3, and 6 months. Neurologic outcome was assessed using the Cerebral Performance Categories (CPC) [13]. The CPC is a five-point scale widely used in assessing neurological outcomes in cardiac arrest. CPC 1 describes good cerebral performance that entails normal neurological function or mild neurological or psychological dysfunction. CPC 2 describes a sufficient cerebral function for independent activity of daily living. CPC 3 depicts severe cerebral disability that ranges from dependence on others for activities of daily living, severe dementia, or paralysis. CPC 4 represents a vegetative state or a coma, and CPC 5 is brain death. CPC 1 (good cerebral performance) and 2 (moderate cerebral disability) were considered favorable neurologic outcomes [14].

### 2.3. Quality Assessment

Two independent investigators assessed the quality of each study. We evaluated the quality of randomized controlled trials using Cochrane’s risk-of-bias tool, assessing risk of bias in randomization, deviations from the study protocol, outcome measurement, selection of the reported result, and bias due to missing outcomes data [15]. We rated the quality of observational studies 0–9 using the Newcastle–Ottawa Scale, with scores based on comparability of groups, quality of outcomes, and cohort selection [16]. High-quality studies have a score ≥7, whereas moderate- and low-quality studies have scores of 4–6 and ≤3, respectively. Differences in study quality assessments were resolved through discussion between researchers.

To assess for heterogeneity, we used Cochrane’s Q statistic (which tests against the null hypothesis that all analyzed studies would share a common effect size) and I^2^ statistic (which determines the percentage of total variance that is expected to occur due to a difference in effect size across studies) [17].

### 2.4. Data Extraction

Data from the included studies were extracted into a standardized Excel spreadsheet (Microsoft Corp). We included patient demographics such as age and comorbidities, initial cardiac rhythm, arrest etiology, the percentage of arrests that were witnessed or treated with bystander CPR, transportation times, and timing to interventions. We recorded survival rates and CPCs at hospital discharge, as well as at 1, 3, and 6 months.

To ensure that the extracted data accurately reflected the information presented in the articles reviewed, data were extracted by two authors independently and compared for any discrepancies. Any discrepancies were resolved through discussion between the two authors.

### 2.5. Statistical Analysis

We used descriptive analyses—mean (standard deviation (SD)) or percentage—to express the extracted data. Where appropriate, we converted median and interquartile range to the mean and SD, as previously described [18]. For continuous variables such as studies’ sample sizes, we inspected the histogram of all studies’ sample sizes first, then categorized them according to their distributions of frequency. Categorical variables for subgroup analyses included World Health Organization (WHO) region, study design (matched cohorts vs. non-matched cohorts vs. randomization), and sample size.

We used random-effects models to compare the outcomes of interest between the pooled populations. We performed random-effects meta-analysis when three or more studies reported the same outcome.

We also performed moderator analyses using categorical variables of studies’ characteristics to identify potential sources of heterogeneity and to compare between subgroups. The degree of heterogeneity was identified once the meta-analysis was performed. For sensitivity analysis, we performed “remove-one study” random-effects meta-analysis to assess the effect of each individual study on the overall effect size. We also used cumulative meta-analysis to assess the efficacy of ECPR compared to CPR over the course of time: we performed random-effects meta-analysis first with the earliest study, then repeated the analysis with the earliest and the second earliest studies, repeated the analysis again using the first three studies, and so on.

Publication bias was evaluated by using Orwin’s Fail-Safe N to predict the number of missing studies or future studies needed to impact the effect size of our primary outcome. Our meta-analysis was performed using the software Comprehensive Meta-Analysis (www.meta-analysis.com, accessed on 13 July 2021).

## 3. Results

### 3.1. Study Selection

Our search identified a total of 2088 studies. In total, 209 full-text articles were reviewed and 13 were included for data extraction (Figure 1). Three studies were prospective [7,18,19], including one randomized controlled trial [7], and 10 were retrospective [7,19,20]. All reported either survival to hospital discharge or neurologic outcome at hospital discharge, or at 1, 3, or 6 months.

### 3.2. Study Quality

All the studies in our meta-analysis were of high quality, with NOS grading scores of ≥7 or a low risk of bias as per Cochrane’s risk-of-bias tool (Table 1).

### 3.3. Summary of Studies

Our meta-analysis included a total of 18,620 patients, 16,701 of whom underwent CPR and 1919 of whom underwent ECPR. Mean age for the CPR group was 61 and mean age for the ECPR group was 56; 67% of patients in the CPR group were male, compared to 85% patients in the ECPR group.

In the CPR group, ventricular tachycardia (VT) or VF was identified in 63% of cardiac arrests, 84% of arrests were witnessed, and 52% of patients received bystander CPR. In studies reporting return of spontaneous circulation (ROSC) among patients who were treated with traditional CPR, ROSC was achieved in 32% of patients [7,21,23,24,27,28,29]. Fifty-seven percent of patients who achieved ROSC received targeted temperature management (TTM) and 33% underwent coronary revascularization via percutaneous catheter intervention or coronary artery bypass grafting (Table 2). In the ECPR group, VT or VF was identified in 61% of arrests, 81% of arrests were witnessed, and 57% received bystander CPR. TTM was performed in 52% of cases, and coronary revascularization in 47% (Table 2).

In terms of primary outcome, good neurologic outcome at hospital discharge was reported in 2 studies [19,23], while 5 studies reported it at 1 month [20,22,25,28,30], 3 studies at 3 months [26,28,30], and 2 studies at 6 months [20,27]; 2 studies reported survival to discharge as their primary outcome [7,21].

Our analysis included six registry-based studies—four based in Japan, one in Paris, France, and one in Seoul, South Korea [20,21,23,25,27,30]. No additional studies from Paris or France were included. Two studies utilized the SAVE-J registry, based in Japan, though they reported different outcomes of interest and were thus both included in this meta-analysis [18,27]. We were unable to identify participating institutions for the national or city-based registries included in this study. None of the Japanese registry-based studies (except for the two SAVE-J studies) overlapped with respect to the years over which data were collected. We included 1 hospital-based study from Japan, though there was also no overlap with respect to the years of data collection between this study and the included registry-based studies from Japan [25]. One hospital-based study from South Korea may have provided some data duplicated in the Korean registry-based study, though we were unable to confirm their participation in the registry [22].

### 3.4. Primary Outcome: Any Favorable Outcome

Our meta-analysis found a benefit of ECPR versus CPR for any favorable outcome (OR 2.84, 95% CI 1.50–5.40, *p* ≤ 0.01) (Figure 2a). The Q statistic of 8 with 12 degrees of freedom (D(f)) with *p* < 0.001 suggests that the effect size identified in our meta-analysis is likely different from the true effect size. The I^2^ was 86%, which suggested that 86% of variance between the effect size among the included studies was due to sampling errors. Cumulative statistics were performed, which showed that the benefit of ECPR over CPR has persisted between 2013 and 2020 (OR 2.84, CI 1.50–5.40, *p* ≤ 0.01) (Appendix A) Publication bias was assessed with Orwin’s Fail-Safe N, which suggested that 4 future studies with a mean OR of 0.4 favoring CPR (similar to Kitada et al. 2020) would be needed to bring the pooled patient population’s OR to 1.0, indicating a similar efficacy between ECPR and CPR.

### 3.5. Sensitivity Analysis

Random-effects meta-analysis with one study removed demonstrated that the benefit of ECPR over CPR for any favorable outcome persisted regardless of the exclusion of any individual study (Appendix A). Our sensitivity analysis demonstrated the robustness of our pooled effect size, and it was not affected by any individual study.

### 3.6. Subgroup Analysis

Our moderator analyses identified a few subgroups with low heterogeneity (Table 3): studies conducted in the Americas and European regions of the WHO had an I^2^ of 0%. Similarly, studies with ≤50 patients were found to have an I^2^ of 0%, while studies with matching cohorts had an I^2^ of 18%. Our meta-analysis found a significant benefit of ECPR versus CPR for our composite outcome of any favorable outcome in studies conducted in the Americas and Western Pacific WHO regions (Table 3); this benefit was not seen in the European WHO region. Similarly, we observed a benefit of ECPR for any favorable outcome in studies with a sample size ≤100 patients but not in those with a sample size >100 patients; the difference between these two groups was significant. Studies that used different designs (matched cohorts, unmatched cohorts, or randomized cohorts) were not significantly different with respect to their observed difference in outcomes associated with ECPR versus CPR.

### 3.7. Secondary Outcomes

#### 3.7.1. Survival to Hospital Discharge

Six studies reported survival to hospital discharge [7,20,21,22,24,26]. Among these studies, there was no difference between survival to hospital discharge in patients treated with ECPR and those treated with CPR (OR 1.68, CI 0.92–3.06 *p* = 0.09) (Figure 3). There was high heterogeneity among these studies (*p*-value for Q statistic <0.001), suggesting that the effect size of our studies is different from the true effect size. Additionally, the I^2^ of 77% for this analysis suggested that 77% of variance between our studies’ effect size and the true effect size was due to sampling errors.

#### 3.7.2. Favorable Neurologic Outcomes

Five studies reported favorable neurologic outcomes at discharge [7,19,21,22,24], 4 at 1 month [23,25,27,30], 5 at 3 months [7,24,26,28,30], and 3 at 6 months [20,27,29]. Our analysis found no significant difference in survival or favorable neurologic outcome at hospital discharge (Figure 4a) or at 1 month (Figure 4b) between patients treated with ECPR and those treated with CPR. We found high heterogeneity among the studies for each of these outcomes.

Treatment with ECPR was associated with higher odds of favorable neurologic outcome at 3 months (OR 5.0, 95% CI 1.90–13.1, *p* < 0.01) (Figure 4c) and at 6 months (OR 4.44, 95% CI 2.3–8.5, *p* < 0.01) (Figure 4d), when compared to CPR. Studies reporting each of these outcomes demonstrated low heterogeneity. For the outcome of favorable neurologic outcome at 3 months, Orwin’s Fail-Safe N analysis suggested that approximately 32 future or missing studies with a mean OR of 0.8 favoring CPR would be needed to negate the benefits of ECPR. Similarly, Orwin’s Fail-Safe N for favorable neurologic outcome at 6 months showed that approximately 21 future or missing studies with a mean OR of 0.8 favoring CPR would be needed to negate the benefits of ECPR.

## 4. Discussion

Our meta-analysis found that ECPR was associated with significantly higher rates of any favorable outcome (a composite outcome of survival to hospital discharge, favorable neurologic outcome at hospital discharge or favorable neurologic outcome at 1, 3, or 6 months after arrest) relative to CPR. We did not observe a significant difference in rates of survival to hospital discharge, or in survival with a favorable neurologic outcome at discharge or at 1 month. ECPR was, however, associated with significantly better CPCs at 3 and 6 months than CPR. This suggests that the benefit associated with ECPR may be primarily in the long term and related to survival with favorable neurologic outcome as opposed to survival alone. There has recently been increased interest in the long-term functional status of survivors of cardiac arrest (such as return to work), suggesting these outcomes are of greater importance to patients, and should be valued by clinicians over short-term outcomes or survival alone [31,32].

A high level of heterogeneity was observed among our studies, though all were judged to be of high quality. Our moderator analyses demonstrated that studies with matched cohorts had lower I^2^ values than the pooled population, which suggest a more consistent effect size among these studies with a more “balanced” patient population. On the other hand, studies with ≤100 patients reported a larger effect size of ECPR than studies with a larger patient population; they also reported low heterogeneity. As a result, we observed a “small study effect” for studies comparing ECPR and CPR. We suggest future studies utilize matched cohorts and larger sample sizes.

Only two studies included in this meta-analysis favored CPR with respect to their primary outcome of interest: Bougouin et al. (2020) and Kitada et al. (2020) [21,25]. Bougouin et al. present the findings of a registry analysis conducted in the Paris region [21]. ECPR was initiated both pre-hospital and on hospital arrival. They demonstrated no significant difference between CPR and ECPR in both matched and unmatched cohorts with respect to the outcome of survival to hospital discharge. Of note, the presence of bystander CPR, an initial shockable rhythm, and a shorter prehospital resuscitation period were all associated with improved outcomes and were all more common in the ECPR cohort of this study, as they were often used by clinicians as guidelines to determine eligibility for ECPR.

Kitada et al., present the results of their propensity score-matched analysis of a registry encompassing 288 critical care centers in Japan [25]. Their outcome of interest was survival with favorable neurologic outcome, and they demonstrated worse outcomes associated with ECPR when compared to CPR for all eligible patients. However, this study included only patients who either underwent ECPR or experienced ROSC and were hospitalized after hospital arrival. Therefore, while all patients treated with ECPR were included in their analysis, those treated with CPR who did not achieve ROSC were excluded; this stands in contrast to most of the studies included in our meta-analysis and may have skewed results in favor of CPR. However, subgroup analysis looking at patients with low or no flow times of greater than 30 or 45 min showed favorable neurologic outcomes in those that received ECPR versus CPR, which is consistent with findings in this study for those subgroups.

### 4.1. Implications for Future Research

Additional research is needed to identify demographic and clinical variables associated with benefit from ECPR. This knowledge would be invaluable to the selection of patients most suited for transfer to an ECPR-capable facility and for ECMO cannulation in OHCA. Additionally, it may have important implications for the design and implementation of ECPR programs within hospital systems and EMS regions.

Additionally, our finding that ECPR was most strongly associated with improvements in long-term neurologic outcomes raises questions regarding its impact on more nuanced functional outcomes, such as eventual discharge home, need for home caregivers, and return to work. Further investigation is needed to qualify the long-term outcomes experienced by patients with OHCA treated with CPR versus ECPR.

Finally, there remains limited evidence regarding the relative costs and benefits associated with ECPR. Prior investigations have noted that ECPR is associated with a net overall benefit in the form of quality adjusted life years, though it is associated with a significant cost, estimated at over $125,000 per patient in the United States [33,34,35]. Additional research is needed to quantify costs and benefits of ECPR compared to CPR across patient populations, and to further investigate the distribution of those costs and benefits across patients, hospitals, payors, and society.

### 4.2. Limitations

Most of the studies included in our analysis were observational; only one small randomized controlled trial was included, which was likely impacted by the “small study effect” based on our analysis. Different studies utilized different outcomes and evaluated those outcomes over different timeframes: perhaps most notably, Kitada et al. did not examine outcomes at 3 or 6 months after arrest, which may have contributed to our finding that ECPR was associated with improved outcomes in those timeframes. Despite the use of propensity score matching by several included studies, the large proportion of observational studies in this meta-analysis limited our ability to fully evaluate for and control confounding variables. For example, some included studies revealed a significant presence of factors known to be associated with favorable neurologic outcome such as younger age after multivariate logistic regression analysis (Schober et al. 2017 [29]) and higher rates of targeted temperature management (Sakamoto et al. 2014 [20]) in the ECPR groups compared to CPR groups. Furthermore, the overall small number of available studies comparing CPR and ECPR for OHCA limited our ability to perform exploratory analysis, such as subgroup analysis or multivariable meta-regression, to identify patient and hospital factors most likely to favor ECPR. Sufficient data were not available at the individual level in many studies on several variables known to affect neurologic outcomes to allow for further analysis including total time with no or low flow state and use of targeted temperature management (Table 1) to help eliminate known confounders associated with long-term favorable neurologic outcomes.

Another important limitation of our study is the potential for duplication of data presented in both registry- and hospital-based studies. Our review of included studies did not demonstrate any significant overlap. Moreover, our sensitivity analysis included a “one-study-removed” analysis, which confirmed that our findings were not significantly altered by the inclusion or removal of any single study. Therefore, while the potential for some duplication of data remains, we are confident that it did not have a significant impact on our findings. The majority of the studies included in this paper obtained the CPC through the analysis of patient data from retrospective chart review instead of direct patient examination, affecting the reliability of the CPC score due to possible bias, and a variability in inter- and intra-reviewer agreement [14]. Lastly, studies that looked at IHCA and ECPR identified variables such as age, Society of Thoracic Surgeons (STS) score, comorbidities, and lab abnormalities as predictors of in-hospital mortality. Not all studies in this paper reported the same set of independent variables; hence, we were unable to perform an exploratory meta-regression to identify predictors of such outcomes [36] due to anticipated insufficient power.

## 5. Conclusions

Our study demonstrated that ECPR was associated with improved CPC at 3 and 6 months following arrest, suggesting its benefit in long-term functional status in OHCA survivors when compared to CPR. Additional research is needed to identify patient demographics and clinical variables associated with benefit from ECPR in OHCA.

## Figures and Tables

**Figure 1 healthcare-10-00591-f001:**
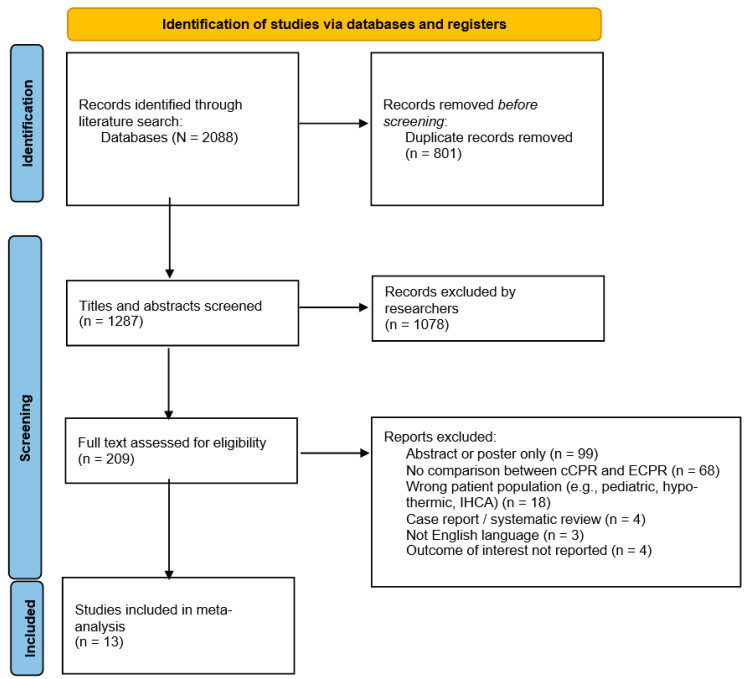
Flow diagram for study selection. Abbreviations: cCPR, conventional cardiopulmonary resuscitation; ECPR, extracorporeal cardiopulmonary resuscitation; IHCA, in-hospital cardiac arrest. Adapted from the PRISMA 2020 statement [10].

**Figure 2 healthcare-10-00591-f002:**
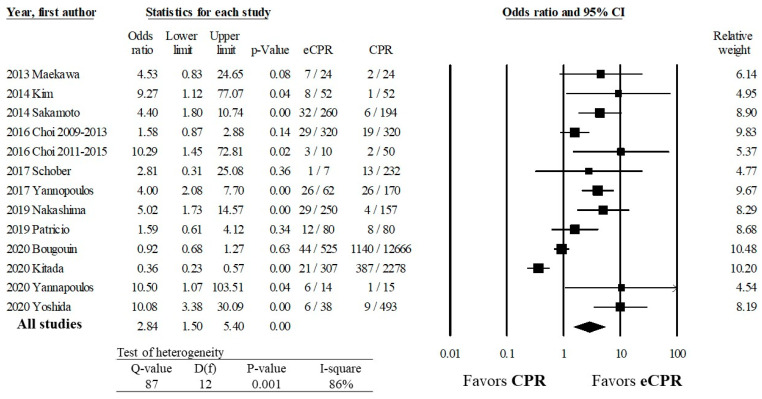
Association of ECPR and conventional CPR with any favorable outcome, defined as survival to hospital discharge or favorable neurologic function at hospital discharge or 30 or more days after cardiac arrest, among patients with OHCA. Abbreviations: CPR, cardiopulmonary resuscitation; ECPR, extracorporeal pulmonary resuscitation; OHCA, out-of-hospital cardiac arrest.

**Figure 3 healthcare-10-00591-f003:**
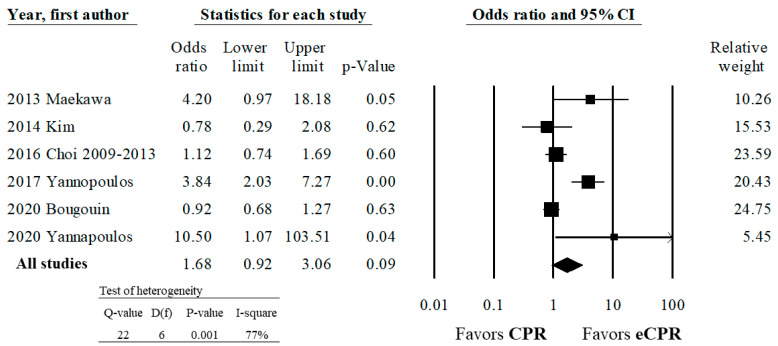
Association of ECPR and conventional CPR with survival to hospital discharge among patients with OHCA.

**Figure 4 healthcare-10-00591-f004:**
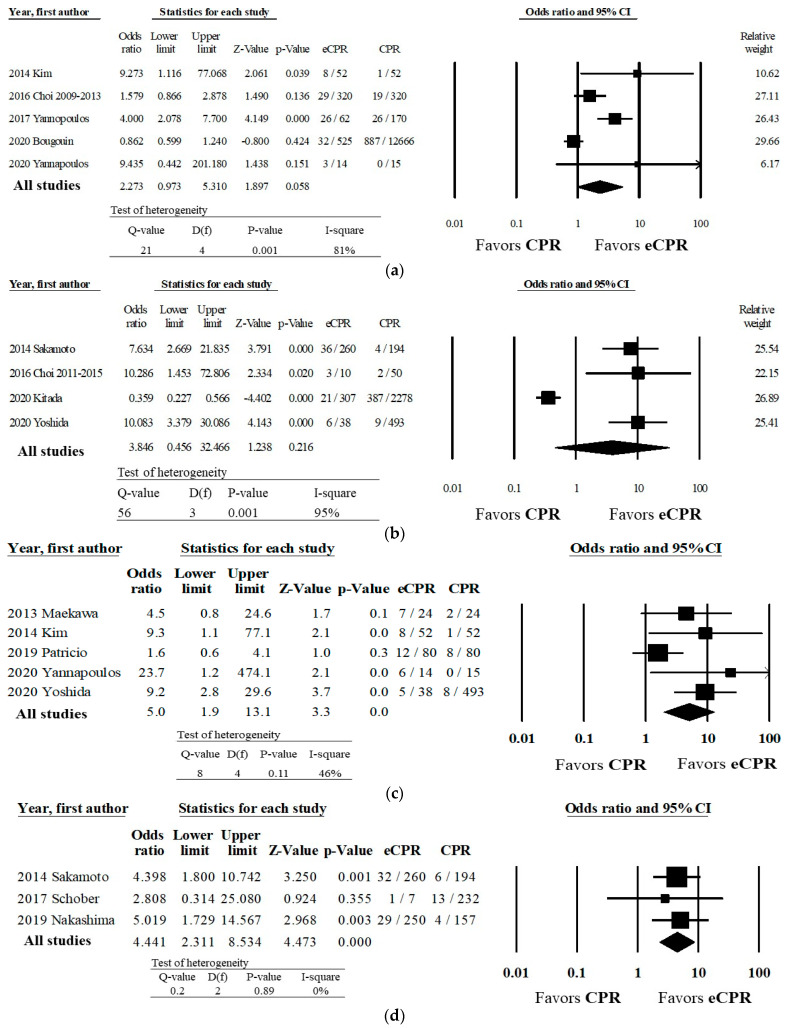
Association of ECPR and conventional CPR with survival with a favorable neurologic outcome among patients with OHCA. Favorable neurologic outcome is defined as Cerebral Performance Category 1 or 2. (**a**) Odds ratio of favorable neurological outcome at hospital discharge; (**b**) Odds ratio of favorable neurological outcome at one month after arrest; (**c**) Odds ratio of favorable neurological outcome at three months after arrest; (**d**) Odds ratio of favorable neurological outcome at six months after arrest.

**Table 1 healthcare-10-00591-t001:** Characteristics, outcome measures, and quality ratings of included studies.

Study	Publication Year	Sample Size (CPR/eCPR)	Location	Study Design	ECMO Cannulation Location ^a^	Primary Outcome	Secondary Outcome(s)	Study Quality Rating
Bougouin et al. [21]	2020	12,666/525	France	Retrospective, Obs	-	Survival to hospital discharge	CPC 1 or 2 at hospital discharge	8
Choi et al. [22]	2016	50/10	Korea	Retrospective, Obs	-	CPC 1 or 2 at 1 month	Survival at 1 month	7
Choi et al. [23]	2016	320/320	Korea	Retrospective, Obs	-	CPC 1 or 2 at hospital discharge	Survival to hospital discharge	8
Kim et al. [24]	2014	52/52	Korea	Retrospective, Obs	ED, Cath Lab	CPC 1 or 2 at 3 months	Survival at 24 h, hospital discharge, and 3 months	8
Kitada et al. [25]	2020	2278/307	Japan	Retrospective, Obs	-	CPC 1 or 2 at 1 month	None	9
Maekawa et al. [26]	2013	24/24	Japan	Retrospective, Obs	-	CPC 1 or 2 at 3 months	None	9
Nakashima et al. [27]	2019	157/250	Japan	Retrospective, Obs	-	CPC 1 or 2 at 6 months	Survival at 6 months	9
Patricio et al. [28]	2019	50/49	Belgium	Retrospective, Obs	-	CPC 1 or 2 at 3 months; survival to ICU discharge *	None	8
Sakamoto et al. [20]	2014	194/260	Japan	Prospective, Obs	-	CPC 1 or 2 at 1 and 6 months	None	7
Schober et al. [29]	2017	232/7	Austria	Retrospective	ED	CPC 1 or 2 at 6 months	None	8
Yannopoulos et al. [19]	2017	170/62	USA	Prospective	Cath Lab	CPC 1 or 2 at discharge	CPC 1 or 2 at 3 months	7
Yannopoulos et al. [7]	2020	15/15	USA	RCT	Cath Lab	Survival to hospital discharge	Survival at 1, 3, and 6 months; CPC 1 or 2 at hospital discharge, 1, 3, 6 months	Low risk
Yoshida et al. [30]	2020	493/38	Japan	Retrospective	-	CPC 1 or 2 at 1, 3 months	Survival at 1, 3 months	7

^a^ (-): Data not reported. * Both listed as primary outcome. Abbreviations: CPR, cardiopulmonary resuscitation; ECPR, extracorporeal cardiopulmonary resuscitation; ECMO, extracorporeal membrane oxygenation; Obs, observational; USA, United States of America; RCT, randomized controlled trial; ED, Emergency Department; Cath Lab, cardiac catheterization laboratory.

**Table 2 healthcare-10-00591-t002:** Demographics and characteristics of arrest in patients treated with conventional CPR and ECPR.

			Past Medical History				Arrest Etiology			Arrest Characteristics						Additional Treatments ^a^	
Study	Age ^b,c^	Male N (%)	DM N (%)	HTN N (%)	HLD N (%)	CAD N (%)	ACS N (%)	PE N (%)	Arrhythmia N (%)	VT/VF N (%)	Witnessed N (%)	Bystander CPR N (%)	Time to Hospital (min) ^b^	Low-Flow Time (min) ^d^	ROSC N (%)	TTM N (%)	CABG, PCI N (%)
Bougouin et al. [21]																	
CPR	66 (16)	8486 (67)	-	-	-	-	196 (37)	18 (3)	-	3167 (25)	9500 (75)	6206 (49)	-	-	4789 (38)	-	966 (20)
ECPR	50 (13)	441 (84)	-	-	-	-	194 (37)	16 (3)	-	357 (68)	509 (97)	425 (81)	-	-	-	-	159 (54)
Choi et al. [22]																	
CPR	59 (12)	38 (76)	-	-	-	-	-	-	-	13 (26)	50 (100)	41 (82)	19 (8)	-	15 (30)	10 (67)	2 (13)
ECPR	58 (6)	7 (70)	-	-	-	-	-	-	-	3 (30)	10 (100)	8 (80)	14 (10)	-	-	6 (60)	5 (56)
Choi et al. [23]																	
CPR ^e^	58 (6)	259 (81)	-	-	-	-	-	-	-	90 (28)	234 (73)	74 (32)	19 (-)	47 (-)	-	34 (11)	-
ECPR	56 (7)	258 (81)	-	-	-	-	-	-	-	93 (29)	227 (71)	96 (30)	19 (-)	54 (-)	-	95 (30)	-
Kim et al. [24]																	
CPR ^e^	55 (8)	38 (73)	6 (12)	12 (23)	-	11 (21)	9 (17)	1 (2)	5 (10)	29 (56)	42 (81)	16 (31)	-	68 (-)	20 (40)	12 (60)	3 (15)
ECPR	53 (8)	40 (77)	11 (21)	13 (25)	-	15 (29)	36 (69)	2 (4)	3 (6)	31 (60)	42 (81)	22 (42)	-	70 (-)	-	14 (27)	29 (56)
Kitada et al. [25]																	
CPR	76 (5)	1457 (64)	-	-	-	-	-	-	-	-	-	1002 (44)	-	-	-	-	-
ECPR	60 (6)	257 (84)	-	-	-	-	-	-	-	215 (70)	-	157 (51)	-	-	-	-	-
Maekawa et al. [26]																	
CPR ^e^	58 (5)	19 (79)	-	-	-	-	-	-	-	14 (58)	24 (100)	14 (58)	28 (3)	52 (-)	-	-	-
ECPR	56 (4)	19 (79)	-	-	-	-	-	-	-	13 (54)	24 (100)	13 (54)	31(3)	51 (-)	-	9 (38)	5 (21)
Nakashima et al. [27]																	
CPR	60 (5)	139 (89)	-	-	-	-	82 (52)	-	-	157 (100)	123 (78)	68 (43)	32 (4)	-	48 (31)	22 (46)	16 (37)
ECPR	58 (5)	227 (91)	-	-	-	-	163 (65)	-	-	250 (100)	183 (73)	115 (46)	32 (4)	55 (5)	-	-	-
Patricio et al. [28]																	
CPR	-	-	-	-	-	-	-	-	-	-	-	-	-	-	26 (52)	-	-
ECPR	-	-	-	-	-	-	-	-	-	-	-	-	-	-	-	-	-
Sakamoto et al. [20]																	
CPR	58 (NR)	172 (89)	-	-	-	-	114 (59)	-	27 (14)	194 (100)	151 (78)	90 (46)	31 (-)	-	-	-	-
ECPR	56 (NR)	235 (90)	-	-	-	-	165 (64)	-	42 (16)	260 (100)	186 (72)	127 (49)	30 (-)	-	-	162 (63)	97 (37)
Schober et al. [29]																	
CPR	60 (6)	173 (75)	44 (19)	67 (29)	-	65 (28)	-	-	-	135 (58)	204 (88)	72 (31)	56 (9)	78 (-)	89 (38)	48 (55)	11 (12)
ECPR	46 (8)	5 (71)	0 (0)	2 (28)	-	1 (14)	-	-	-	4 (57)	6 (86)	2 (28)	42 (11)	93 (-)	-	3 (43)	2 (28)
Yannopoulos et al. 2017 [19]																	
CPR	56 (7)	124 (73)	37 (22)	63 (37)	54 (32)	24 (14)	-	-	-	170 (100)	131 (77)	128 (75)	-	-	-	-	-
ECPR	58 (10)	44 (71)	12 (19)	30 (48)	22 (36)	6 (9)	-	-	-	62 (100)	50 (80)	52 (84)	-	-	-	-	46 (74)
Yannopoulos et al. 2020 [7]																	
CPR	58 (11)	11 (73)	3 (20)	5 (33)	2 (13)	4 (27)	-	-	-	15 (100)	13 (87)	12 (80)	50 (-)	-	2 (13)	2 (100)	2 (100)
ECPR	59 (10)	14 (93)	3 (20)	2 (13)	1 (7)	2 (13)	-	-	-	15 (100)	11 (73)	13 (87)	48 (-)	59 (-)	-	15 (100)	-
Yoshida et al. [30]																	
CPR	72 (16)	307 (62)	-	-	-	-	20 (4)	10 (2)	-	0 (0)	-	-	16 (4)	-	-	-	-
ECPR	61 (16)	27 (71)	-	-	-	-	8 (21)	10 (26)	-	0 (0)	-	-	11 (5)	39 (6)	-	-	-

^a^ for patients treated with CPR, reported as percent of patients who achieved ROSC; ^b^ continuous variables are represented as the mean (SD); ^c^ (-) denotes data not reported or not applicable; ^d^ “low flow time” represents total time of chest compressions and time to compressions when time to compressions was provided. For patients treated with conventional CPR, the endpoint of low flow time is determined by ROSC or time of death. For patients treated with ECPR, the endpoint of low flow time is the initiation of ECMO. ^e^ Data from propensity score-matched cohorts presented. Abbreviations: DM, diabetes mellitus; HTN, hypertension; HLD, hyperlipidemia; CAD, coronary artery disease; ACS, acute coronary syndrome; PE, pulmonary embolism; VT/VF, ventricular tachycardia/ventricular fibrillation; CPR, cardiopulmonary resuscitation; ECMO, extracorporeal membrane oxygenation; ROSC, return of spontaneous circulation; TTM, targeted temperature management; CABG, coronary artery bypass grafting; PCI, percutaneous coronary intervention; ECPR, extracorporeal cardiopulmonary resuscitation.

**Table 3 healthcare-10-00591-t003:** Moderator analyses using studies’ characteristics as categorical variables.

Moderator Variables		Number of Studies	Odds Ratio (95% CI)	*p*-Value	Q-Value	D(f)	*p*-Value	I²	Between-Group Comparison *p*-Value
WHO region	AMR	2	4.3 (2.3–8.1)	0.001	0.63	1	0.43	0%	0.001
	EURO	3	0.99 (0.74–1.3)	0.97	2	2	0.37	0%	
	WPR	8	3.5 (1.2–9.9)	0.02	67	7	0.001	90%	
Sample size of ECPR group	<50 patients	5	7.5 (3.6–15)	0.001	2	4	0.81	0%	0.014
	51–100 patients	3	3.1 (1.4–6.9)	0.005	3	2	0.18	43%	
	>100 patients	5	1.5 (0.7–3.3)	0.34	41	4	0.001	90%	
Categories of patient analysis	Matched	4	2.0 (1.1–3.6)	0.019	3.6	3	0.3	18%	0.36
	Unmatched	8	2.7 (1.1–6.6)	0.24	77	7	0.001	91%	
	Randomized	1	10.5 (1.06–100+)	0.044	NA	NA	NA	NA	

Abbreviations: WHO regions, World Health Organization regions; AMR, Americas region; EURO, European region; NA, not applicable; WPR, Western Pacific region; ECPR, extracorporeal cardiopulmonary resuscitation.

## Data Availability

Not applicable.

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
