# Peer review of "A Comparison between Conventional and Extracorporeal Cardiopulmonary Resuscitation in Out-of-Hospital Cardiac Arrest: A Systematic Review and Meta-Analysis"

_healthcare, 2022, doi:10.3390/healthcare10030591_

Round 1

Reviewer 1 Report

Dear Authors

Thank you for allowing me to review the manuscript entitled ‘’

 A Comparison of Conventional and Extracorporeal Cardiopulmonary Resuscitation in Out-of-Hospital Cardiac  Arrest: A Systematic Review and Meta-Analysis’’.

The main goal of the study is to analyse the survival  and neurologic outcomes among patients treated with ECPR or CPR in OHCA.

The main outcome is that ECPR was associated with improved CPC at 3 and 6 months following arrest.

I congratulate the authors for the manuscript.

I have some questions and concerns:

  • Please define ECPR inside the manuscript and not only in the abstract.
  • Clinical studies have shown that >70years, female, insulin-dependent diabetes, severe pulmonary hypertension, STS score >35, type/A aortic dissection, aortic cross-clamp time >150 min and pre-ECLS blood lactate >15 mmol/L as risk factors of in-hospital mortality. Instead coronary artery disease, intra-aortic balloon pump implantation, ECLS start in the operating room, and transapical left ventricular venting, were associated with a better outcome in clinical studies (1). I suggest the authors to analyze the outcome from their studies regarding these influencing factors.
  • Can the authors please clarify the use of left ventricular venting in ECLS in the studies that they choose. These is recommended in the guidelines but not many centers do that. Therefore, analyzing the impact of left ventricular venting/unloading on survival is beneficial.

References

  • Bonacchi M et al. Outcomes' predictors in Post-Cardiac Surgery Extracorporeal Life Support. An observational prospective cohort study. Int J Surg.2020 Oct;82:56-63.doi: 10.1016/j.ijsu.2020.07.063.

Thank you

Author Response

I have some questions and concerns:

  • Please define ECPR inside the manuscript and not only in the abstract.

We thank the reviewer for pointing this out. ECPR is defined in the Introduction of the manuscript, but we can make it more clear to the reader by modifying the sentence to:

“The advent of venoarterial extracorporeal membrane oxygenation (VA ECMO) has allowed for continued treatment and lifesaving attempts in patients with refractory cardiac arrest [6]. Extracorporeal cardiopulmonary resuscitation (ECPR) is defined as the use of VA ECMO in the context of ongoing refractory cardiac arrest.”

  • Clinical studies have shown that >70years, female, insulin-dependent diabetes, severe pulmonary hypertension, STS score >35, type/A aortic dissection, aortic cross-clamp time >150 min and pre-ECLS blood lactate >15 mmol/L as risk factors of in-hospital mortality. Instead coronary artery disease, intra-aortic balloon pump implantation, ECLS start in the operating room, and transapical left ventricular venting, were associated with a better outcome in clinical studies. I suggest the authors to analyze the outcome from their studies regarding these influencing factors.

We thank the reviewer for this comment. We planned to use meta-regressions to evaluate the association between important continuous independent variables and the efficacy of ECPR versus CPR. However, the small number of studies and the amount of studies not reporting those important variables, prevented us from performing a meta-regression. For example, the studies included in our analysis did not report the incidence of pulmonary hypertension, aortic dissection, aortic cross-clamp time, or all the required variables to calculate an STS score. Furthermore, only four studies included ECMO cannulation location (23, 28, 29, 7), and only four studies included intra-aortic balloon pump implantation (7, 25, 26, 29). Therefore, we did not perform a meta-regression using these independent variables due to anticipated insufficient power.

We included a statement under Limitations to report this: “We did not perform an exploratory meta-regression to measure the association between important independent variables such as serum lactate levels, aortic dissection, time of aortic cross-clamp, etc., and the efficacy of ECPR versus CPR, due to insufficient power.”

  • Can the authors please clarify the use of left ventricular venting in ECLS in the studies that they choose. These is recommended in the guidelines but not many centers do that. Therefore, analyzing the impact of left ventricular venting/unloading on survival is beneficial.

We thank the reviewer for this interesting suggestion. Four studies reported the use of a left ventricular venting device (7, 25, 26, 29), which was an intra-aortic balloon pump, but only two studies performed a subgroup analysis in this group of patients and their outcome of interest was favorable neurologic outcome and not survival. Maekawa et al found no significant difference in favorable neurologic outcome between patients who received ECPR plus IABP versus ECPR alone (p=0.051). Nakashima et al did not find a significant effect of ECPR plus IABP in patient’s neurologic outcome (-1.73, -4.27 to 1.09, for sustained VF/pVT and -1.51, -4.46 to 1.11, for PEA/asystole). Therefore, this is an intervention that will need further studies to evaluate its efficacy.

References:

Maekawa K, Tanno K, Hase M, Mori K, Asai Y. Extracorporeal Cardiopulmonary Resuscitation forPatients With Out-of-Hospital Cardiac Arrest of Cardiac Origin: A Propensity-Matched Study andPredictor Analysis*. Crit Care Med. 2013;41(5):1186-1196. doi:10.1097/CCM.0b013e31827ca4c8

Nakashima T, Noguchi T, Tahara Y, et al. Patients With Refractory Out-of-Cardiac Arrest andSustained Ventricular Fibrillation as Candidates for Extracorporeal Cardiopulmonary Resuscitation -Prospective Multi-Center Observational Study. Circ J Off J Jpn Circ Soc. 2019;83(5):1011-1018.doi:10.1253/circj.CJ-18-1257

Reviewer 2 Report

The research is well done.
The methodology is correct, and the data processing is also adequate
There's nothing to change about that 

just a minor suggestion: The research is about comparing between two methods of resuscitation, perhaps the best title could be

"A Comparison between Conventional and Extracorporeal  Cardiopulmonary Resuscitation in Out-of-Hospital Cardiac  Arrest: A Systematic Review and Meta-Analysis"

-Strength:
-The objectives set by the authors are developed correctly
-The length of the manuscript is sufficient
-The research carried out is a relevant and little studied topic

Weaknesses:
-I think the number of tables are excessive
-Researchers can replace a table (example table 3) with a figure that summarizes the same (example circles or bars)
The research leaves the message that extracorporeal cardiopulmonary resuscitation (ECPR) is more effective than CPR in the management of refractory out-of-hospital cardiac arrest. This could be interpreted as not encouraging CPR
Could the researchers mention which are the most favorable scenarios to promote the ECPR vs CPR resuscitation technique?

Finally, researchers could leave a message about the importance of providing CPR in cases of attending cardiopulmonary resuscitation and how a rescuer could select candidates for ECPR.

Author Response

  • Just a minor suggestion: The research is about comparing between two methods of resuscitation, perhaps the best title could be

"A Comparison between Conventional and Extracorporeal  Cardiopulmonary Resuscitation in Out-of-Hospital Cardiac  Arrest: A Systematic Review and Meta-Analysis"

We thank the reviewer for this suggestion. The title has been changed to "A Comparison between Conventional and Extracorporeal Cardiopulmonary Resuscitation in Out-of-Hospital Cardiac Arrest: A Systematic Review and Meta-Analysis"

  • Weaknesses:
    -I think the number of tables are excessive:

We thank the reviewer for this suggestion. Figure 2B (Sensitivity analysis for composite outcome of any favorable outcome)and 2C (Cumulative statistical analysis for composite outcome) have been removed from the manuscript and will be included under supplementary data.

  • The research leaves the message that extracorporeal cardiopulmonary resuscitation (ECPR) is more effective than CPR in the management of refractory out-of-hospital cardiac arrest. This could be interpreted as not encouraging CPR. Could the researchers mention which are the most favorable scenarios to promote the ECPR vs CPR resuscitation technique? Finally, researchers could leave a message about the importance of providing CPR in cases of attending cardiopulmonary resuscitation and how a rescuer could select candidates for ECPR.

We thank the reviewer for these helpful comments. The following changes were implemented in the Introduction:

The recent ARREST trial—a small, phase 2, single-center, open-label, adaptive, safety and efficacy randomized controlled trial (RCT),and the only RCT to date directly comparing ECPR and CPR, found a higher survival rate in the ECPR group (43%) in comparison to the CPR group (7%) [7]. This study utilized rigorous inclusion and exclusion criteria; patients needed to have shockable rhythm, no ROSC after three defibrillation shocks, and an estimated transfer time of less than 30 minutes. These criteria ensured that patients had minimal intervals of no-flow time. The strict exclusion criteria of trauma and presence of multiple comorbidities, also aimed to exclude patients with low likelihood of survival. We observed that strict selection criteria for ECPR can be associated with improved patient outcomes compared to CPR. Despite this promising evidence, the ARREST trial consisted of a small sample size with highly trained personnel, and it is unclear if their result can be generalized outside this setting and population.ECPR should not replace conventional CPR until there are more clear and definitive criteria for ECPR; prompt and high-quality CPR remains the main effort to optimize patients’ outcomes.

Round 2

Reviewer 1 Report

Dear Authors

Thank you for allowing me to re-review this manuscript.

Thank you also for replying to all my questions and concerns.

I have no further comments.